# Compliance with smoke-free legislation in public places: An observational study in a northeast city of Bangladesh

**Saifur Rahman Chowdhury**[1,2]*, **Tachlima Chowdhury Sunna**[1], **Dipak Chandra Das**[1], **Mahfuzur Rahman Chowdhury**[3], **H. M. Miraz Mahmud**[4☯], **Ahmed Hossain**[5,6☯]

1 Department of Public Health, North South University, Dhaka, Bangladesh, 2 Department of Health Research Methods, Evidence, and Impact (HEI), McMaster University, Hamilton, Ontario, Canada, 3 Department of English, North East University Bangladesh, Sylhet, Bangladesh, 4 Bangladesh Center for Communication Programs (BCCP), Dhaka, Bangladesh, 5 Health Services Administration, College of Health Sciences, University of Sharjah, Sharjah, United Arab Emirates, 6 NSU Global Health Institute (NGHI), North South University, Dhaka, Bangladesh

☯ These authors contributed equally to this work.
* saifur@mcmaster.ca, saifur.rahm1994@gmail.com

**Data Availability Statement:** All relevant data are within the manuscript and its Supporting information files.

## Abstract

### Background

Bangladesh is one of the highest tobacco-consuming countries in the world, with a large number of adult users of a variety of smoked and/or smokeless tobacco products. Bangladesh tobacco control act prohibits smoking in public places and requires the owners of public places to display 'no smoking' signages.

### Objectives

The objective of this study was to assess the level of compliance with the tobacco control act (smoke-free laws) in public places in a northeast city of Bangladesh.

### Methods

This cross-sectional study was conducted between June 1 and August 25, 2020, across 673 public places in Sylhet city, Bangladesh. The data was collected using a structured observational checklist that included variables such as the presence of active smoking, the presence of designated smoking areas, the display of 'no smoking' signages, evidence of recent smoking such as ashes, butts/bidi ends, and the presence of smoking aids.

### Results

Among 673 public places, a total of 635 indoor locations and 313 outdoor locations were observed. Only 70 (11%) indoor locations were found to be in good compliance, and 388 (61.1%) indoor locations were found to be in moderate compliance with smoke-free laws. On the other hand, only 5 (1.6%) outdoor locations were in good compliance, and 63 (20.1%) outdoor locations were in moderate compliance with smoke-free laws. The overall compliance with smoke-free laws at indoor locations was 52.7%, and at outdoor locations

**Funding:** Support for this study was provided by the Bangladesh Center for Communication Programs (BCCP) with funding awarded by Bloomberg Philanthropies to Johns Hopkins University. Grant number is GC#BCCP/Tobacco Control/2020-57. Saifur Rahman Chowdhury received this grant in reference to the technical and cost proposal from the Bangladesh Center for Communication Programs (https://www.bangladesh-ccp.org/). The content of this publication is solely the responsibility of the authors and do not necessarily represent the official views of Bloomberg Philanthropies or Johns Hopkins University. The funders had no role in study design, data collection and analysis, decision to publish, or preparation of the manuscript.

**Competing interests:** The authors have declared that no competing interests exist.

was 26.5%. The highest compliance was observed at healthcare facilities (58.6%) and the least at transit points (35.7%) for indoor locations. In outdoor locations, the highest compliance was observed at offices and workplaces (37.1%) and the least at transit points (2.2%). Higher active smoking was observed in public places where there was an absence of 'no smoking' signage and the presence of points of sale (POSs) (p-value <0.05). Further, higher active smoking was observed in places where any smoking aids, cigarette butts, bidi ends, or ashes were present (p-value <0.05).

## Conclusion

This study found moderate compliance at indoor locations and very low compliance at outdoor locations. The government should focus more on implementing smoke-free laws in all kinds of public places, particularly at most frequently visited places and transit sites. 'No smoking' signages should be displayed per legislation across all public places. Policymakers should consider the prohibition of POS in/around a public place as it has a positive effect on smoking.

## 1. Introduction

Tobacco use is one of the most preventable causes of premature death in the world, contributing to 6 million deaths every year [1]. Smoking and second-hand smoke (SHS) are collectively serious and growing public health concerns globally, with a large number of tobacco-associated preventable deaths occurring in lower-income countries [2]. The number of tobacco smokers is increasing rapidly because of the availability of cheap tobacco products, the lack of strong tobacco control regulations, and the weak enforcement of existing regulations. It has been estimated that nearly a third of the world's population, those over the age of 15 years, smokes cigarettes [3], and smoking prevalence is on the rise, especially in developing countries [4]. Bangladesh is one of the top ten countries in the world, with a high current smoking prevalence of 35.3% [5]. Future projections suggest that tobacco smoking will kill more than 8 million people each year worldwide by the year 2030, with 80% of these premature deaths occurring in low and middle-income countries [6].

All non-smokers are potentially in danger of exposure to SHS because of the smokers [6]. Exposure to SHS is now unequivocally proven to be as harmful as active smoking, causing death, disease, and disability. Every year, exposure to SHS causes over 880,000 premature deaths worldwide [7, 8]. There is a relatively high prevalence of exposure to SHS in Bangladesh [5]. According to a nationwide survey conducted in Bangladesh, the prevalence of SHS was 43% [9]. The Global Burden of Disease Study estimates that in 2019, tobacco was responsible for around 157,862 fatalities in Bangladesh, which is approximately 19% of all deaths [10]. According to World Health Organization (WHO), each year in Bangladesh, 1.2 million people suffer from diseases that are associated with tobacco use [11]. A study conducted in Bangladesh revealed that 61,000 children were suffering from diseases due to exposure to SHS in 2018 [12]. A ten-year prospective study that included twenty thousand adult participants found that smoking was responsible for 25% of deaths among males and 7.6% of deaths among females [13]. These pieces of evidence demonstrate that Bangladesh has a huge number of deaths and illnesses related to tobacco use, necessitating national attention to this massive issue. When it comes to protecting the general public from SHS, the emphasis has been placed

on the enforcement of appropriate legislation throughout the world. A Cochrane Review of 50 studies from developed countries shows that enforcing laws can reduce exposure to SHS, especially in workplaces and public places [14].

Bangladesh is distinguished as the first signatory of the World Health Organization (WHO) Framework Convention on Tobacco Control (WHO FCTC), which was ratified on May 10, 2004. In Bangladesh, the Smoking and Tobacco Products Usage (Control) Act was acted in 2005 and amended in 2013. Bangladesh tobacco control act requires the prohibition of smoking in public places and owners of public places to display 'no smoking' signages. In the context of developing countries, Bangladesh's experiences in enforcing public health laws have been dismal. However, effective implementation of the legislation necessitates ongoing monitoring to ensure that legal provisions are being followed and that decisions can be used for midcourse correction [15, 16]. Several studies have been conducted in Bangladesh to assess the prevalence and patterns of tobacco use, but no studies measured the observance of multiple compliance indicators and smoking in public places [17–21].

Smokers' behavior is influenced in part by their understanding of smoke-free legislation. There has been evidence of the relationship between the effective implementation of legislation regarding smoking restrictions in public places and the reduction of smoking behavior [22–24]. Chapman et al. estimated the contribution of smoke-free workplaces to the declines in cigarette consumption in Australia and the USA. They reported that smoke-free workplaces are responsible for an annual reduction of 602 million cigarettes [25]. A study conducted in Spain by Jimenez-Ruiz found that the prevalence of exposure to environmental tobacco smoke decreased from 49.5% in 2005 to 37.9% in 2007 (a 22% reduction) after the implementation of smoke-free laws in the country [26]. According to Wakefield, a study conducted in the United States found that smoking restrictions and smoking bans in public places may help to reduce teenage smoking [27].

However, there have been many challenges faced in Bangladesh in implementing the Smoking and Tobacco Products Usage (Control) Act. To the best of our knowledge, there has been no study in Bangladesh as of yet to measure compliance with smoke-free legislation. In these circumstances, assessing compliance with the tobacco control act in public places is of great importance to determine the extent to which the law is being implemented. Therefore, we aimed to measure the status of compliance with legal provisions that protect the public against the harms of SHS exposure in a northeast city of Bangladesh, where the findings of the study would help to identify the potential areas of violations and inform policymakers for strengthening enforcement measures.

## 2. Materials and methods

### 2.1 Study design

This was a cross-sectional study conducted in Sylhet City, Bangladesh. The study type was an observational field study. The study was conducted between June 1, 2020, and August 25, 2020.

### 2.2 Study area

There are three major cities in Bangladesh. These are Dhaka, Chattogram, and Sylhet cities. Among these, Sylhet city has been conveniently selected for this study because of its status as a significant metropolitan city in the country, which is located in the northeast of the country. The study area was the municipal areas of Sylhet City Corporation. Its total area is 26.5 sq km. It comprises 27 wards and 207 mahallas (areas). This is a city with multi-dimensional culture and a variety of places and people. Its total population is 475,138 [28].

**Table 1. Designated public places used for assessing smoke-free legislation compliance and the specific areas where the assessments were carried out\*.**

| Category of public place | Specific areas within the public place for assessing smoke-free compliance |
|---|---|
| **Accommodation facility** | Reception, lounge, at least two rooms on different floors, lobby areas, one toilet, at least one backside corridor (if any), bar (if any), restaurant (if any), poolside area (if any) [a] |
| **Eatery** | The entire facility, including the toilets |
| **Office and workplace** | Reception, common waiting room, at least two office rooms, employee retiring or common room (if any), one toilet, meeting room, lobby (if any), at least one backside corridor or balcony (if any), canteen (if any) [a] |
| **Healthcare facility** | Reception, at least one male and one female ward (where applicable), one office room, one doctor's duty room, one toilet, one patients' waiting area, canteen (if any) [a] |
| **Transit point** | Main entrance area, central core area, at least two public toilets, information area, and waiting area (where applicable) [a] |
| **Most frequently visited other public place** | Main entrance area, central core area, at least two public toilets, information area, and waiting area (where applicable) [a] |

\*Adapted from the existing Guide for Conducting Compliance Studies with Smoke-free Law [30]

[a]: All of these areas (if applicable) needed to be assessed for smoke-free compliance in the facility.

## 2.3 Study sites

Study sites included all the public places except for the educational institutions as defined in the Smoking and Tobacco Products Usage (Control) (Amendment) Act, 2013 within Sylhet City [29]. These include–

1. **Accommodation facilities**: any lodging service paid for on a short-term basis, including residential hotels, motels, and rest houses.

2. **Eateries**: restaurants and cafeterias surrounded by walls on all sides.

3. **Offices and workplaces**: government office, semi-government office, autonomous office, private office, court building, library, indoor workplace.

4. **Healthcare facilities**: public/private hospitals/clinics.

5. **Transit points**: airport building, seaport building, river port building, railway station building, bus terminal building, designated queues or places for passengers waiting to ride on public transport.

6. **Most frequently visited other public places**: cinema hall, exhibition center, theatre hall, fitness center, sports facility, shopping center, public toilet, children park, fairs, any other public area to be combinedly used by the general people or, any or all places declared time to time by the government or local government organization by general or special order.

The designated public places that were used for the assessment of smoke-free legislation compliance and the specific areas where the assessments were carried out are outlined in Table 1.

## 2.4 Operational definitions

**Compliance**: It is the degree to which the Smoking and Tobacco Products Usage (Control) (Amendment) Act, 2013, Bangladesh is being obeyed.

**Smoke-free legislation**: The law that prohibits smoking in a specified area i.e., the Smoking and Tobacco Products Usage (Control) (Amendment) Act, 2013, Bangladesh.

**Public Places**: All the places that are defined as public places in the Smoking and Tobacco Products Usage (Control) (Amendment) Act, 2013, Bangladesh.

**Smoking**: Inhaling or exhaling the smoke of tobacco and also includes keeping or controlling any flamed tobacco products.

**Active smoking**: Active smoking in a public place was marked as present if anyone was seen smoking during the researcher's visit at the public place being observed for the study.

**Designated smoking area (DSA)**: The section of a public place that has been set apart and designated as a smoking area by the owner, caretaker, manager, or other individual in charge of the public place in which smoking is permitted. DSA was marked as present if it was noticed during the researcher's visit at the public place being observed for the study.

**'No smoking' signage**: According to the Smoking and Tobacco Products Usage (Control) (Amendment) Act, 2013, every public place shall arrange to display a caution notice **"Refrain from Smoking, It is a Punishable Offence"** in Bengali and in English language at the entrance and in one or more places inside a public place. The size of a caution notice board in a public place shall be at least 40 centimeters × 20 centimeters. Every public place shall arrange to display the caution notice in red letters against a white background or in white letters against a red background with a no-smoking sign.

**Cigarette buts, bidi ends, or ashes**: Cigarette buts, bidi ends, or ashes were marked as present as evidence of previous smoking in a particular public place if any of these was noticed during the researcher's visit at the public place being observed for the study.

**Smoking aids**: According to the Smoking and Tobacco Products Usage (Control) (Amendment) Act, 2013, no smoking aids such as ashtrays, ashbins, matchboxes, or lighters can be kept in the smoke-free area.

## 2.5 Sample size and sampling technique

In accordance with existing literature, the sample size for this study has been calculated [30]. As there is no evidence of proportion from the previous studies, so, based on the expected compliance rate of 50%, a margin of error of 5%, design effect of 2, and anticipating a 10% loss to observe, a sample of 673 public places was calculated by using Epi Info Version 7.2. All the samples were proportionately distributed among all types of public places in Sylhet City to get the maximum sample from each type of public place.

Sylhet City Corporation has a database of all types of places under its jurisdiction, including restaurants and workplaces (private and public). A list of all types of public places was collected from Sylhet City Corporation. Then, using simple random sampling, 10 wards were selected from the list of 27 wards in Sylhet City Corporation. Finally, a simple random sampling method was adopted among each type of public place in the selected wards to obtain the desired sample size.

## 2.6 Study tool

A structured observational checklist was used to record the findings. It was developed based on an existing guide on 'Assessing compliance with smoke-free law' [30], reviewing relevant literature and the Smoking and Tobacco Products Usage (Control) (Amendment) Act, 2013, Bangladesh. The guide on 'Assessing compliance with smoke-free law' was developed as a part of a collaborative effort between the Campaign for Tobacco-Free Kids, Johns Hopkins Bloomberg School of Public Health and International Union against Tuberculosis and Lung Disease (IUALTD), 2014 [30]. The IUALTD/Johns Hopkins developed the tool for WHO as a valid

tool for assessing the compliance of smoke-free laws. To observe the indoor and outdoor of distinguished public places, the checklist included the indicators like the absence of active smoking, absence of designated smoking area (if not permitted), display of 'no smoking' signages, display of 'no smoking' signages at the main entrance and other conspicuous places, 'no smoking' signage complies with the law, absence of cigarettes buts, bidi ends or ashes, and the absence of smoking aids. We made some adjustments to the existing compliance study tool [30]. For example, the indicators "display of 'no smoking' signages at the main entrance and other conspicuous places and 'no smoking' signage complies with the law" was taken because these are part of the Smoking and Tobacco Products Usage (Control) (Amendment) Act, 2013, Bangladesh. Additionally, the indicator "absence of cigarettes buts, bidi ends or ashes" was considered based on similar studies as evidence of recent smoking. The questionnaire was piloted, and required modification was brought in. The questionnaire is attached in S1 File. This checklist has been used in previous studies conducted in India and Nepal, and it is easy to use [22, 24, 31, 32].

### 2.7 Data collection

Four teams consisting of two members (two data collectors) in each team collected data. They were trained by Principal Investigator on smoke-free law and its provisions, along with filling of the standard checklist used for the study. On-site training was also provided before data collection to maintain the quality of data collection. Trained field data collectors visited each of the sampled public places on weekdays at an unannounced time to capture typical behavior. Prior to data collection, they obtained the informed consent of the individuals in charge of the public places. Except for the person in charge of the institute, there was no interaction with anyone else in the sampled public place. The teams visited the government buildings during the office timings (9:00 to 17:00). In healthcare facilities, visits were done from 11:00 to 12:00 and 16:00 to 20:00 and also during visiting hours. The transit points, shopping malls, bars, and restaurants were visited during the busiest hours (evening hours). Depending on the area covered, the average time spent at each public place was 45 minutes to one hour.

### 2.8 Quality control

To avoid the possibility of personal bias, all observations were carried out by data collectors who had been trained in filling out the observation checklist. The Principal Investigator, along with co-principal investigators, visited 10% of the sampled facilities and independently cross-checked their findings against those obtained by data collectors, using the same observational checklist that data collectors used. The visit was carried out on the same day in order to ensure the robustness of the monitoring. The findings were found to be 100% consistent between the data collectors and the principal investigator.

### 2.9 Statistical analysis

All the data were screened for any missing value. After screening, the data were entered and coded in MS Access and analyzed using the statistical software SPSS, version 22. Descriptive statistics were presented through tables. The compliance with specific indicators of smoke-free legislation in different public places, both indoors and outdoors, was presented separately in tables. For indoors, a location was given a rating of 'good compliance', 'moderate compliance' or 'poor compliance" depending on the number of indicators it met: 5–7, 3–4, or 0–2 for good compliance, moderate compliance, or poor compliance, respectively. For outdoors, a location was leveled as 'good compliance', 'moderate compliance', or 'poor compliance' if the place

complied with 5–6 indicators, 3–4 indicators, or 0–2 indicators, respectively. In order to calculate 'total/overall compliance', the values of 'individual compliance indicators' were added together and divided by the total number of indicators. This method was applied in previous research to find the total/overall compliance [22, 24, 32]. Compliance status with different smoke-free indicators was presented by bar diagrams. Chi-squared/Fisher's exact test was performed to show the significance of the association of active smoking in public places with different compliance indicators and other predictors. All statistical analyses were two-sided, and a $p$-value $<0.05$ was considered statistically significant.

### 2.10 Ethical issues

The survey protocol was reviewed and approved by the North South University Research Ethics Committee (NSU-IRB-2020/54). In public places with restricted entry (like hospitals, hotel rooms, and offices), verbal and prior informed consent was taken from the in-charge. Data privacy and confidentiality were maintained, and data were used only for research purposes.

## 3. Results

### 3.1 Description of the observed public places

A total of 673 public places were observed in this study, and the distribution of specific types of public places can be found in the S1 Table in S2 File. Among the observed places, some were only indoor locations, some were only outdoor locations, and some were both indoor and outdoor locations. Thus, the observation of 673 places covered 635 indoor locations and 313 outdoor locations. Indoor locations included 58 accommodation facilities, 176 eateries, 147 offices and workplaces, 90 healthcare facilities, 150 most frequently visited places, and 14 transit points. Out of 313 outdoor locations, the number of accommodations facilities, eateries, offices and workplaces, healthcare facilities, most frequently visited places, and transit points were 8, 53, 110, 47, 64, and 31, respectively (Table 2).

**Table 2. Types of observed public places.**

| Type of location | n (%) |
|---|:---:|
| **Indoors** | |
| Accommodation facilities | 58 (9.1) |
| Eateries | 176 (27.7) |
| Offices and workplaces | 147 (23.2) |
| Healthcare facilities | 90 (14.2) |
| Most frequently visited places | 150 (23.6) |
| Transit points | 14 (2.2) |
| **Total** | **635 (100.0)** |
| **Outdoors** | |
| Accommodation facilities | 8 (2.6) |
| Eateries | 53 (16.9) |
| Offices and workplaces | 110 (35.1) |
| Healthcare facilities | 47 (15.0) |
| Most frequently visited places | 64 (20.5) |
| Transit points | 31 (9.9) |
| **Total** | **313 (100.0)** |

**Table 3. Compliance with specific indicators of smoke-free legislation in different public places in Sylhet City (observation indoors, n = 635).**

| Compliance indicators | All public places, n (%) |
|---|---|
| Absence of active smoking | 558 (87.9) |
| Absence of a designated smoking area indoors (where not permitted) [a] | 224 (99.6) |
| Presence of 'no smoking' signage | 141 (22.2) |
| Display of 'no smoking' signage at the main entrance and other conspicuous places | 34 (5.4) |
| 'No smoking' signage complies with the law | 9 (1.4) |
| Absence of cigarette buts, bidi ends, or ashes | 408 (64.3) |
| Absence of smoking aids such as ashtrays, ashbins, matchboxes, lighters | 561 (88.3) |
| **Good compliance** | 70 (11.0) |
| **Moderate compliance** | 388 (61.1) |
| **Poor compliance** | 177 (27.9) |
| **Total compliance\* (%)** | **52.7** |

[a]: n = 225

Good compliance: Compliance with 5–7 indicators; Moderate compliance: Compliance with 3–4 indicators; Poor compliance: Compliance with 0–2 indicators

\* Total compliance was calculated by averaging the percentages of various compliance indicators.

## 3.2 Overall compliance with smoke-free legislation

**3.2.1 Compliance at indoors of different public places.** Compliance with specific indicators of smoke-free legislation at indoors of different public places is described in Table 3. Only 70 (11%) indoor locations were found to be in good compliance, and 388 (61.1%) indoor locations were found to be in moderate compliance with smoke-free laws. The total average compliance with specific indicators of smoke-free legislation at indoor locations was 52.7%.

In terms of average compliance, the transit points scored least than all other public places, which was only 35.7%. The highest average compliance was observed in healthcare facilities (58.6%), followed by eateries (58.1%), offices and workplaces (57.0%), accommodation facilities (45.7%), and most frequently visited other places (39.2%) (Fig 1).

**3.2.2 Compliance at outdoors of different public places.** Table 4 describes the compliance with specific indicators of smoke-free legislation at outdoors of different public places. Only 5 (1.6%) outdoor locations were in good compliance, and 63 (20.1%) outdoor locations were in moderate compliance with smoke-free laws. Overall compliance of outdoor locations was 26.5%.

In outdoor locations, the offices and workplaces scored the highest average compliance than all other public places, which was 37.1%. The least compliance was observed in transit points which was only 2.2%. In terms of compliance of other places, the most frequently visited places' average compliance was 14.6%, accommodation facilities' was 20.8%, eateries' was 27.4%, and healthcare facilities' was 32.6% (Fig 2).

## 3.3 Compliance with different indicators of smoke-free legislation

**3.3.1 Compliance at indoors of different public places.** In Fig 3, compliance with different indicators of smoke-free legislation at indoors of different public places is described. Smokers were observed in 77 (12.1%) indoor locations of different public places. Among the indoor locations, 124 smokers were observed in the 57 (38.0%) most frequently visited places that are the venue with the highest number of smokers. Only 141 (22.2%) indoor places were observed where the owners displayed 'no smoking' signage. Compliance of 'no smoking' signages with

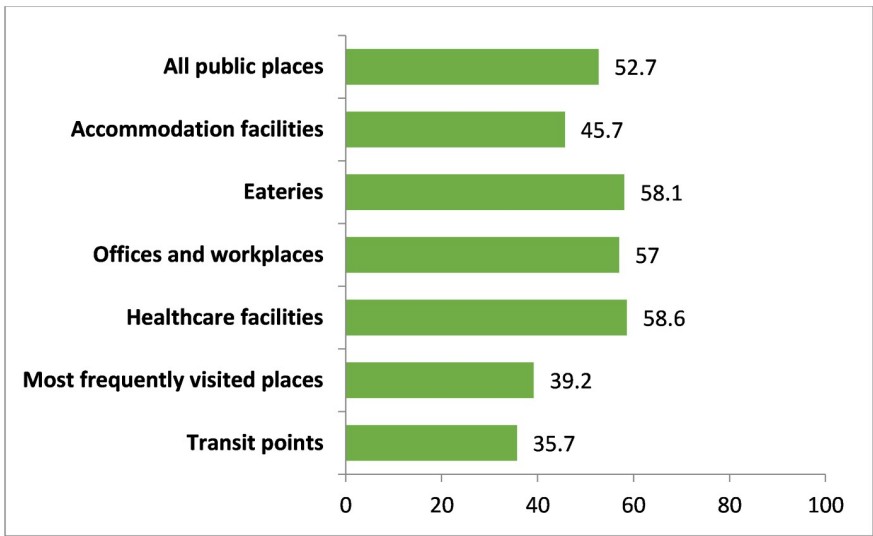

**Fig 1. Overall compliance with smoke-free legislation (indoor).** Note: All public places (n = 635), Accommodation facilities (n = 58), Eateries (n = 176), Offices and workplaces (n = 147), Healthcare facilities (n = 90), Most frequently visited places (n = 150), Transit points (n = 14).

the "Smoking and Tobacco Products Usage (Control) (Amendment) Act" at indoor places is described in S4 Table in S2 File. Among 141 indoor places, where the owners displayed 'no smoking' signage, 34 (24.1%) displayed signages were in both the main entrance and other conspicuous places. Regarding whether the signage in indoor locations conforms with the legislation, just 9 (6.4%) of the exhibited signages were deemed to be entirely compliant.

A significant number of indoor places, 227 (35.7%), were found with any cigarette buts, bidi ends, or ashes during the observation. Fig 3 also represents the distribution of the presence of smoking aids in different indoor places. The number of indoor places where smoking aids were found is 74 (11.7%). The name of different types of smoking aids that were present at indoors of public places is stated in the S6 Table in S2 File.

**Table 4. Compliance with specific indicators of smoke-free legislation in different public places in Sylhet City (observation outdoors, n = 313).**

| Compliance indicators | All public places, n (%) |
|---|---|
| Absence of active smoking | 195 (62.3) |
| Presence of 'no smoking' signage | 33 (10.5) |
| Display of 'no smoking' signage at the main entrance and other conspicuous places | 12 (3.8) |
| 'No smoking' signage complies with the law | 7 (2.2) |
| Absence of cigarette buts, bidi ends, or ashes | 57 (18.2) |
| Absence of smoking aids such as ashtrays, ashbins, matchboxes, lighters | 193 (61.7) |
| **Good compliance** | 5 (1.6) |
| **Moderate compliance** | 63 (20.1) |
| **Poor compliance** | 245 (78.3) |
| **Total compliance* (%)** | **26.5** |

Good compliance: Compliance with 5–6 indicators; Moderate compliance: Compliance with 3–4 indicators; Poor compliance: Compliance with 0–2 indicators

* Total compliance was calculated by averaging the percentages of various compliance indicators.

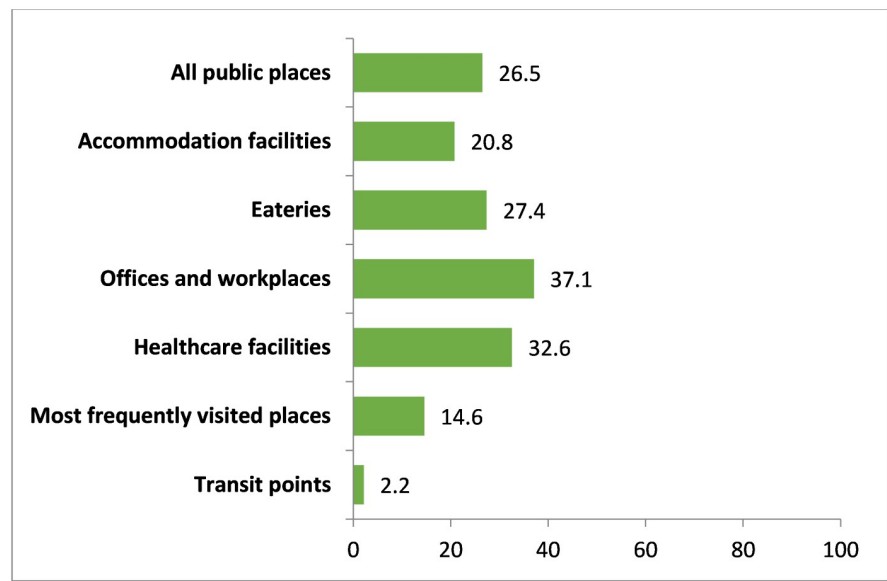

**Fig 2. Overall compliance with smoke-free legislation (outdoor).** Note: All public places (n = 313), Accommodation facilities (n = 8), Eateries (n = 53), Offices and workplaces (n = 110), Healthcare facilities (n = 47), Most frequently visited places (n = 64), Transit points (n = 31).

**3.3.2 Compliance at outdoors of different public places.** Fig 4 describes the compliance with different indicators of smoke-free legislation at outdoors of different public places. Smokers were observed in 118 (37.7%) outdoor locations of different public places. Among outdoors, the highest total number of smokers (n = 187) was present at 29 (93.5%) transit points, followed by a total of 125 persons at 44 (68.7%) most frequently visited places. In the case of outdoor places, 33 (10.5%) places were observed with 'no smoking' signage display. Compliance of 'no smoking' signages with the "Smoking and Tobacco Products Usage (Control) (Amendment) Act" at outdoor places is described in the S5 Table in S2 File. Among the outdoor places with 'no smoking' signage displayed, 12 (36.4%) were observed with signage displayed in both the main entrance and other conspicuous places. Concerning the compliance of the 'no smoking' signage with the tobacco control act at outdoor places, only 3 (9.1%) of the places' signages complied with the law perfectly.

Most of the outdoor places, 256 (81.8%), were found with the presence of any cigarette buts, bidi ends, or ashes during observation. The number of outdoor places where smoking aids were observed is 120 (38.3%). The name of different types of smoking aids that were present at outdoors of public places is given in the S7 Table in S2 File.

## 3.4 Availability of points of sale (POSs)

In terms of availability of points of sale (POSs), 91 (29.1%) of the public places had tobacco shops inside their location boundary (Fig 5). Almost in all public places, tobacco POSs (641 (95.2%)) were found within 100 meters of the observed venue (Fig 6).

## 3.5 Presence of active smoking in different public places

In this study, comparatively more places were observed with active smoking (54.1%), where there was the presence of any smoking aids in indoor locations (p-value <0.001). Similarly,

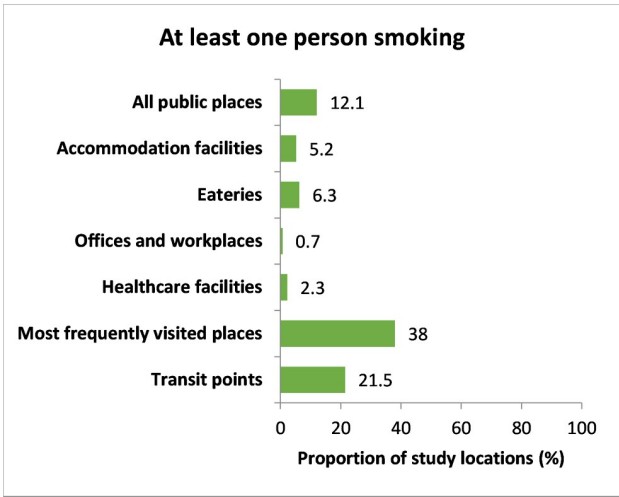

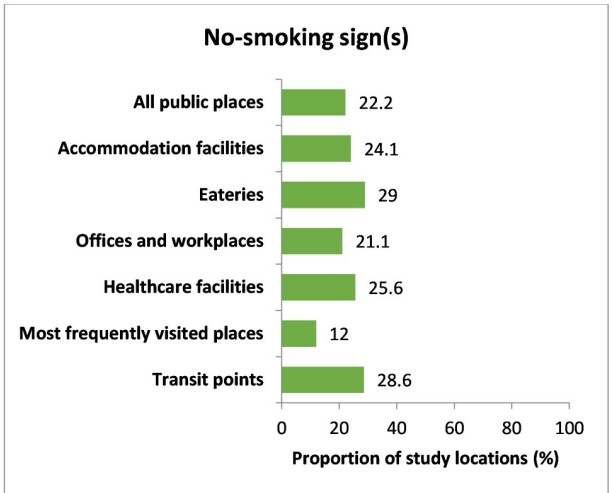

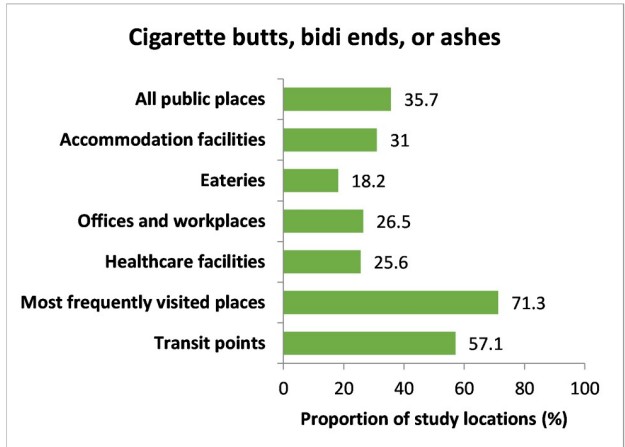

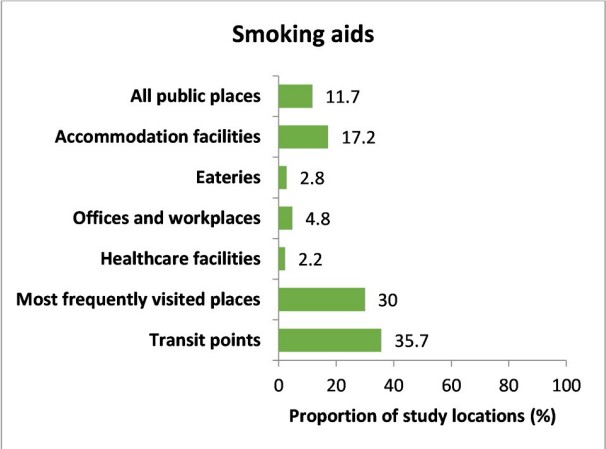

**Fig 3. Indoor observation of smoking, no-smoking sign(s), cigarette buts, bidi ends or ashes, and smoking aids.** Note: All public places (n = 635), Accommodation facilities (n = 58), Eateries (n = 176), Offices and workplaces (n = 147), Healthcare facilities (n = 90), Most frequently visited places (n = 150), Transit points (n = 14).

active smoking was comparatively more prevalent (36.1%) in indoor public places where there was the availability of any POSs within the venue (p-value <0.001). The observance of places with active smoking (32.2%) was also associated with the presence of cigarette buts, bidi ends, or ashes in indoor places (p-value <0.001). Active smoking was comparatively less prevalent (7.1%) in indoor public places where there was a display of 'no smoking' signages (p-value 0.038). Active smoking was more prevalent (38.0%) in the most frequently visited places (p-value <0.001) (Table 5).

In the case of outdoor places, most of the places with active smoking (81.3%) were observed where POSs were available within the venue/location (p-value <0.001). Same as indoor places, the presence of smoking aids was associated with a greater number of places with active smoking (70.8%) in outdoor places (p-value <0.001). The presence of cigarette buts, bidi ends, or ashes, availability of POSs within 100 meters of the venue/location, and type of location were also associated with smoking behavior at outdoors of different public places (p-value <0.05) (Table 6).

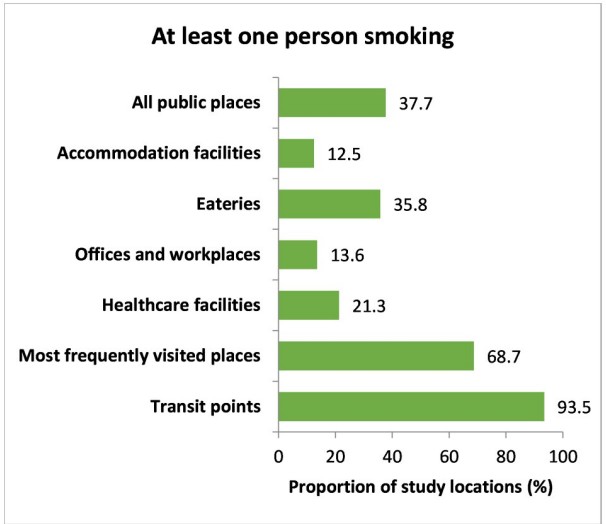

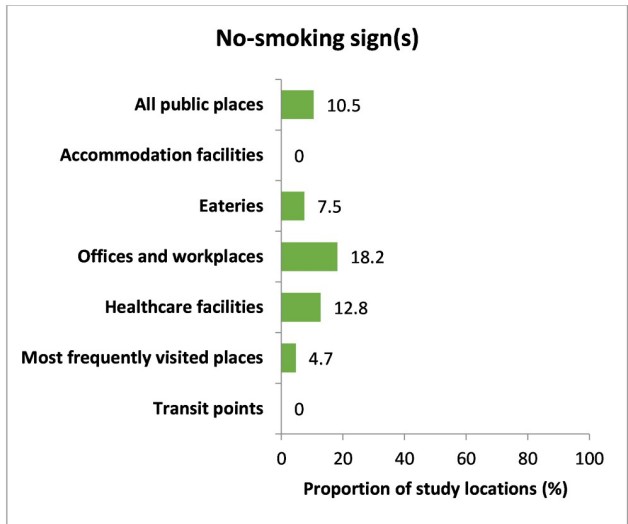

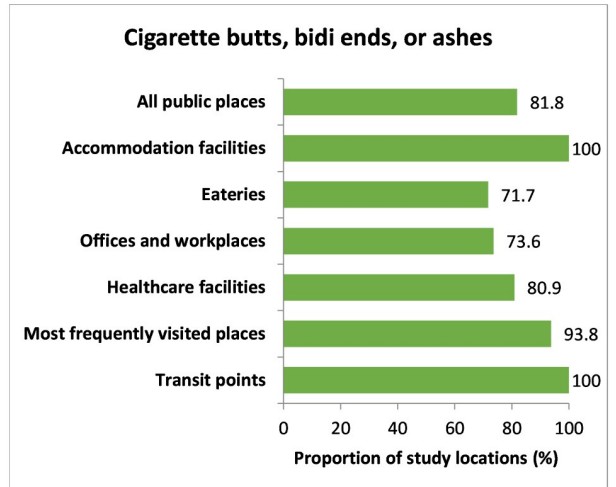

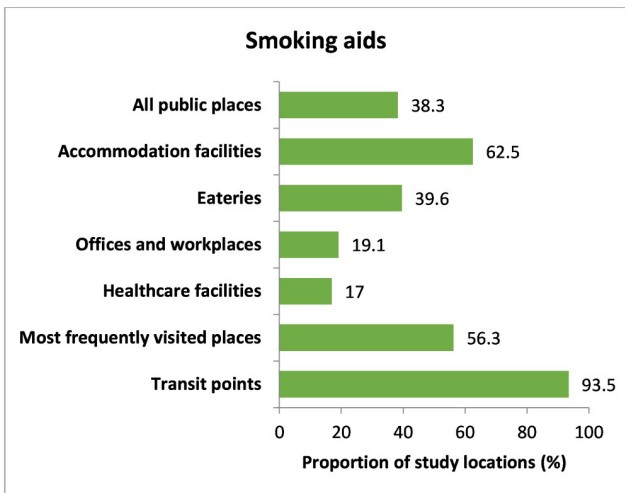

**Fig 4. Outdoor observation of smoking, no-smoking sign(s), cigarette buts, bidi ends or ashes, and smoking aids.** Note: All public places (n = 313), Accommodation facilities (n = 8), Eateries (n = 53), Offices and workplaces (n = 110), Healthcare facilities (n = 47), Most frequently visited places (n = 64), Transit points (n = 31).

## 4. Discussion

This was the first study to investigate the compliance of smoke-free legislation using a standard checklist in public places in Sylhet City, Bangladesh. The aspect of our research was measuring overall compliance to smoke-free legislation in public places, comprised of an assessment of various parameters, such as observing active smoking, display of 'no smoking' signages, and proxy evidence of active tobacco usage. The results revealed that the average compliance with specific indicators of smoke-free legislation at indoor locations was 52.7%, and at outdoor locations was 26.5%. In previous studies, wherein compliance monitoring to smoke-free legislation in four jurisdictions of India, Sikkim state, Vilupuram district and Coimbatore city in Tamil Nadu, and Shimla city in Himachal Pradesh using a similar study tool reported compliance rates varying from 82% to 100% [33]. Goel et al. conducted a study in one of the districts of Punjab in India in the year 2010 to assess compliance with smoke-free legislation with a

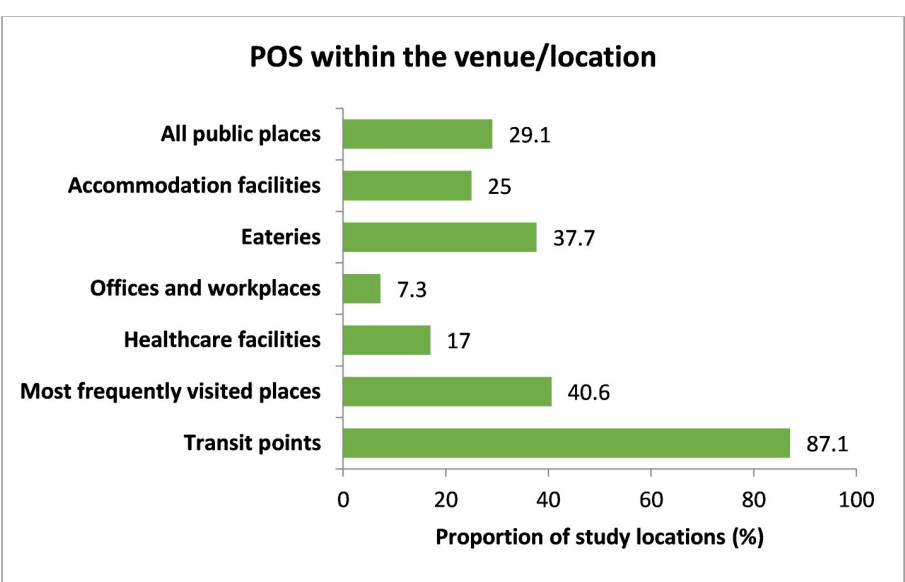

**Fig 5. Availability of POS within the venue/location (inside boundary).** Note: All public places (n = 313), Accommodation facilities (n = 8), Eateries (n = 53), Offices and workplaces (n = 110), Healthcare facilities (n = 47), Most frequently visited places (n = 64), Transit points (n = 31).

similar checklist and found that the overall compliance was 83.8% [22]. In another study conducted in a district of North India, the authors reported that the overall compliance with smoke-free legislation was 92.3% [24]. In a study, wherein compliance surveys in 38 jurisdictions across India were recorded; the authors reported that 51% of the sites demonstrated full compliance with smoke-free laws [31]. A study conducted in 2019 in the Biratnagar metropolitan city in Province 1 of Nepal revealed that the overall compliance with smoke-free legislation

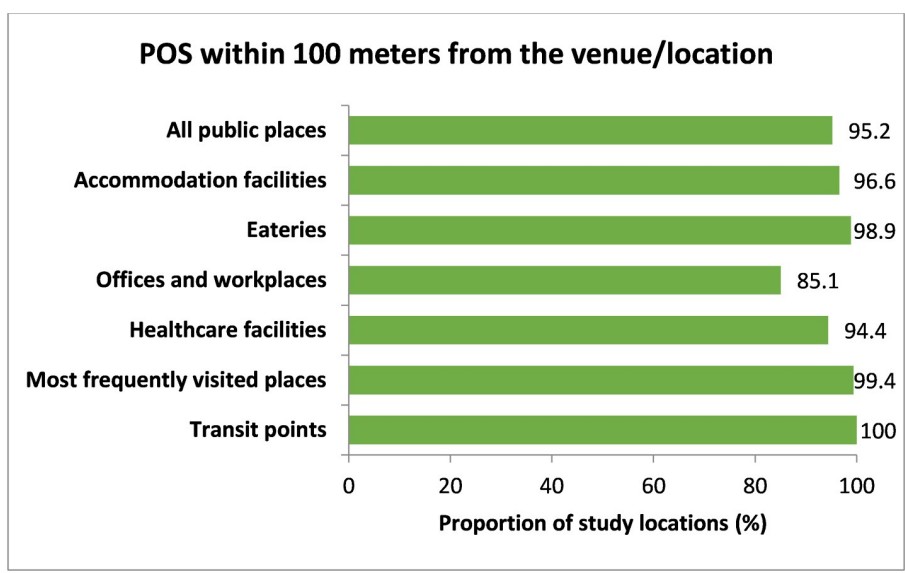

**Fig 6. Availability of POS within 100 meters from the venue/location.** Note: All public places (n = 673), Accommodation facilities (n = 58), Eateries (n = 177), Offices and workplaces (n = 148), Healthcare facilities (n = 90), Most frequently visited places (n = 166), Transit points (n = 34).

**Table 5. Smoking behavior in different public places (indoor, n = 635).**

| Variables | Smoking | | p-value [a] |
|---|---|---|---|
| | No, n (%) | Yes, n (%) | |
| **Designated smoking area indoors (n = 410)** | | | |
| No | 318 (84.6) | 58 (15.4) | 0.095 |
| Yes | 25 (73.5) | 9 (26.5) | |
| **Signage display** | | | |
| No | 427 (86.4) | 67 (13.6) | **0.038** |
| Yes | 131 (92.9) | 10 (7.1) | |
| **Presence of cigarette buts, bidi ends, or ashes** | | | |
| No | 404 (99.0) | 4 (1.0) | **<0.001** |
| Yes | 154 (67.8) | 73 (32.2) | |
| **Presence of smoking aids (ashtrays, ashbins, matchboxes)** | | | |
| No | 524 (93.4) | 37 (6.6) | **<0.001** |
| Yes | 34 (45.9) | 40 (54.1) | |
| **Availability of POS within the venue/location (n = 278)** | | | |
| No | 206 (94.9) | 11 (5.1) | **<0.001** |
| Yes | 39 (63.9) | 22 (36.1) | |
| **Availability of POS within 100 meters of the venue/location** | | | |
| No | 31 (96.9) | 1 (3.1) | **0.161** |
| Yes | 527 (87.4) | 76 (12.6) | |
| **Type of location** | | | |
| Accommodation facilities | 55 (94.8) | 3 (5.2) | **<0.001** |
| Eateries | 165 (93.8) | 11 (6.3) | |
| Offices and workplaces | 146 (99.3) | 1 (0.7) | |
| Healthcare facilities | 88 (97.8) | 2 (2.2) | |
| Most frequently visited places | 93 (62.0) | 57 (38.0) | |
| Transit points | 11 (78.6) | 3 (21.4) | |

Note:

[a] p-value from the chi-square test; if a cell value is less than five, then from the Fisher's Exact test.

was 56.4% [32]. Similarly, a study conducted in 2019 in Karachi, Pakistan, showed almost identical results, where compliance with smoke-free indicators was 57% [34]. The difference in our study is that we assessed the compliance level separately for indoor and outdoor locations. Both indoors and outdoors, it was found that only a small number of public places had a good level of compliance, while the majority of public places had a poor level of compliance with the laws. Similar findings were reported in a study conducted in South Bengaluru, India, where only 1.9% of the observed public places showed full compliance, and 28.1% of the observed public places showed partial compliance with the laws [35].

In our study, active smokers were observed in 77 (12.1%) indoor locations and 117 (37.7%) outdoor locations of different public places. In a previous study in North India, only 6% of the observed public places were found with people actively smoking [24]. Kaur et al. studied educational institutes and restaurants in a city in southern India (Chennai) and found that active smoking was evident at 15% of the sites [36]. The study conducted in the Biratnagar metropolitan city in Province 1 of Nepal found that active smoking was present in 44.8% of all public places [32]. Moreover, the study conducted in Karachi, Pakistan, found active smoking in 24% of observed public places [34]. The successful implementation of smoke-free policy can be

**Table 6. Smoking behavior in different public places (outdoor, n = 313).**

| Variables | Smoking | | p-value [a] |
|---|---|---|---|
| | No, n (%) | Yes, n (%) | |
| **Signage display** | | | |
| No | 172 (61.4) | 108 (38.6) | 0.354 |
| Yes | 23 (69.7) | 10 (30.3) | |
| **Presence of cigarette buts, bidi ends, or ashes** | | | |
| No | 56 (98.2) | 1 (1.8) | <**0.001** |
| Yes | 139 (54.3) | 117 (45.7) | |
| **Presence of smoking aids (ashtrays, ashbins, matchboxes)** | | | |
| No | 160 (82.9) | 33 (17.1) | <**0.001** |
| Yes | 35 (29.2) | 85 (70.8) | |
| **Availability POS within the venue/location** | | | |
| No | 178 (80.2) | 44 (19.8) | <**0.001** |
| Yes | 17 (18.7) | 74 (81.3) | |
| **Availability of POS within 100 meters of the venue/location** | | | |
| No | 15 (88.2) | 2 (11.8) | **0.036** |
| Yes | 180 (60.8) | 116 (39.2) | |
| **Type of location** | | | |
| Accommodation facilities | 7 (87.5) | 1 (12.5) | <**0.001** |
| Eateries | 34 (64.2) | 19 (35.8) | |
| Offices and workplaces | 95 (86.4) | 15 (13.6) | |
| Healthcare facilities | 37 (78.7) | 10 (21.3) | |
| Most frequently visited places | 20 (31.3) | 44 (68.8) | |
| Transit points | 2 (6.5) | 29 (93.5) | |

Note:

[a] p-value from the chi-square test; if a cell value is less than five, then from the Fisher's Exact test.

attributed to multiple factors like vigorous enforcement of smoke-free legislation in Bangladesh and the involvement of multiple stakeholders that could reduce the prevalence of active smoking in public places.

The present study's findings showed that transit sites and most frequently visited places had very high violations of nearly all indicators of the legislation. This is not surprising and has been reported in various studies. A study conducted in the Alwar district of Rajasthan, India, reported that educational institutions and healthcare facilities performed well, while restaurants and transit points performed poorly [37]. In the previous study done by Goel et al. in the district of Punjab in 2010, a similar finding of poor compliance with smoke-free laws in transit sites and restaurants, bars, and shopping malls was reported [24]. Similarly, the study conducted in the Biratnagar metropolitan city in Province 1 of Nepal reported that lower compliance was observed in eateries, entertainment, hospitality, and shopping venues (26.3%) and public transportation and transit (43%) [32]. This suggests that policymakers and implementers need to focus on the implementation of legislation at transit sites and most frequently visited places where the majority of people are exposed to second-hand smoke.

Our study showed that there was a lower prevalence of active smoking in indoor public places where there was a display of 'no smoking' signage. This may be due to the fact that such signages may be more noticed by people, and the perceived idea of being caught and fined increases the likelihood of compliance. The study in the Biratnagar metropolitan city in

Province 1 of Nepal reported that there was a higher likelihood of active smoking where there was an absence of 'no smoking' notice [32]. Bonfill et al. in Spain reported that the presence of appropriate signages prohibiting smoking is associated with a much higher likelihood of compliance with smoke-free laws [38]. Similarly, Apsley et al. in Scotland observed the deterrent effect of smoke-free legislation, including the display of signages on reducing second-hand smoke levels [39]. However, the study in Punjab state, North India, and a study in Greece observed that signage was not a strong determinant of smoking behavior [22, 40].

Our data indicated that the presence of cigarette buts, bidi ends, or ashes and the presence of smoking aids (ashtrays, matchboxes, and lighters) were significantly associated with a higher number of places with active smokers. Similar findings were reported in previous studies [22, 23]. It can be explained that smoking aids could positively influence smokers to smoke in a respective public place.

An interesting finding in the present study is that public places with more availability of POSs within/around the location were significantly associated with a higher prevalence of people with active smoking. Previous studies also reported that the availability of shops selling tobacco products acts as a predictor of active smoking [23].

When viewed from the perspective of a developing nation such as Bangladesh, the overall barriers to the successful implementation of tobacco control policies could include a lack of inter-sectorial coordination, a shortage of resources, low penalization of violators, a lack of adequate and continuous monitoring and implementation of smoke-free laws, and most importantly, low awareness among the general public regarding the health impact of tobacco and the laws that are related to it. According to the level of compliance that was found in our research, it is evident that there is a gap between policy action and public health gains. The variable rates of compliance in different types of public places may suggest that the factors that contributed to the success of one type of public place may be applicable to the success of another type of public place. However, in order to achieve a higher degree of compliance, it is required to employ certain tactics that are tailored to the nature of the public place in question. According to the findings of a study carried out in India, various facilitators and barriers to implementing smoke-free laws are present in particular types of public places [41]. It has been stated that public awareness of the law and strong support from the government in implementation, including the imposition of fines, are factors that facilitate the implementation of policies [41]. This has also been reported in a study in Nigeria [42]. Additionally, a series of sensitization programs for authorized officers and persons in charge of compliance, as well as public health campaigns to raise awareness of the public health benefits of the smoke-free law, are more likely to smoothen enforcement, reduce resistance among stakeholders, and eventually ensure compliance. A 'tobacco awareness week' or 'tobacco awareness fortnight' could be observed to raise public awareness. During these events, the general public would be educated about the policy and provisions, and a strict approach would be taken, in which the general public would be immediately fined, sending a message that the government is serious about the law [43]. Moreover, there should be a regular evaluation of the specific measures employed for smoke-free public areas at various locations, which may have a positive effect.

Studies conducted in India and Nepal reported that people were more likely to smoke if the owners/managers of the public places themselves smoked [22, 32]. Various measures can be taken to discourage smoking by the owners/managers, which ultimately lead to a reduction in active smoking. According to the Smoking and Tobacco Products Usage (Control) (Amendment) Act, 2013, Bangladesh, the manager/owner of the institution is held accountable for the instances where the laws are violated. The managers and owners should be well-trained and updated about the laws regarding smoking in public places, including the fines and punishment levied in the cases where these laws are breached.

Along with the robust implementation of the law, it is also necessary to make the public aware of the existence of such policies [41–43]. The accountable department can use mainstream media and social platforms to make sure people know about the laws of smoking in public places. To reach the younger generations earlier, textbooks should be updated, and mandated classes should be well adapted to change the social norm.

The protection of public health requires a number of different interventions, one of which is a complete smoke-free policy. However, merely enforcing the legislation is not enough. If a smoke-free law is not accompanied by a robust enforcement system that has clearly defined regulations, then the law will not have the desired effect of lowering the number of people who use tobacco products and who are exposed to SHS [44, 45]. As a result, the government of Bangladesh needs to move quickly in order to set up a coordinated enforcement system that will facilitate people to comply with the smoke-free laws. A combination of both strong legislative implementations to enhance compliance and improvement in social norms that deem smoking in public places harmful can contribute significantly to reducing second-hand smoke exposure.

### 4.1 Strengths and limitations

This is the first study of its kind conducted in Bangladesh to assess the status of compliance with specific indicators of the tobacco control act in public places in Sylhet City, Bangladesh. This study included all types of public places except for educational institutions as defined in the Smoking and Tobacco Products Usage (Control) (Amendment) Act, 2013. The studied places were randomly chosen, thus diminishing selection bias.

However, this study has some limitations. First, this study is limited by its cross-sectional design. Second, the study has been conducted in a selected city in Bangladesh, so the generalizability of the results across the country is limited. Third, two investigators visited public places and captured compliance using a standard checklist. Though recording air nicotine levels in public places would have been a better method to record compliance, it may not be possible for a low–middle-income country like Bangladesh with resource constraints.

## 5. Conclusions and recommendations

This observational cross-sectional study assessed the status of compliance with the tobacco control act in public places. This study found moderate compliance at indoor locations and very low compliance at outdoor locations. The highest overall compliance at indoors was observed in healthcare facilities and least in transit points, while at outdoors, the highest overall compliance was observed in offices and workplaces and least in transit points. A high frequency of active smoking was observed at outdoors of observed public places, including healthcare facilities, and a significant number of active smokers was also observed in indoor public places. Among all indoor places, most of the owners of public places did not display any 'no smoking' signage. However, the majority of the displayed 'no smoking signage' did not comply with the law properly. Indirect indicators of smoking, such as the presence of cigarette buts, bidi ends, or ashes were also observed in a notable number of public places. A significant number of public places were also not free from the presence of smoking aids. Points of sale (POSs) were observed at a high rate within the boundary of the locations, and it was present at around 100 meters of almost all public places. Smoking was more prevalent in a public place, where there was an absence of 'no smoking' signage and the presence of cigarette buds, bidi ends, or ashes, smoking aids, and POSs inside of the location or around it.

Based on the findings of our study, we recommend that government should focus more on implementing smoke-free laws in all kinds of public places, particularly at transit sites and

most frequently visited places. 'No smoking' signages should be displayed per legislation across all public places. Outdoors within the boundary of public places should also be included under the definition of 'public place', particularly for healthcare facilities and other significant public places. POSs should be prohibited in public places, particularly in/around 100 meters of healthcare facilities and other significant public places, as it has a positive effect on smoking. Future studies should be conducted across the country with a large sample size to generalize the results. Recording air nicotine levels in public places would have been a better method to record compliance; future studies should assess air nicotine levels. Regular follow-up studies should be conducted to know the extent of compliance.

## Supporting information

**S1 File. Data collection form.**
(DOCX)

**S2 File. Supplementary tables.**
(DOCX)

**S1 Dataset.**
(SAV)

## Acknowledgments

The authors are very much grateful to Mohammad Shamimul Islam, Team Leader, BCCP Tobacco Control Program, Bangladesh Center for Communication Programs (BCCP), for his continuous supervision, kind cooperation, and encouragement during the entire period of this research project. Heartiest thanks also to Mohammad Shahjahan, Director & CEO, Bangladesh Center for Communication Programs (BCCP), for his continuous cooperation in successfully completing this project. The authors would like to put forward utmost respect to the authorities of all public places in Sylhet City for their consent and support in collecting data. The authors are much grateful to all the data collectors (Sumon Ahmad, Tokee Tahmid, Md. Shamsul Hoque, Sushangkar Debnath, Mahmudur Rahman Chowdhury, Md Ikbal Hossain, Md. Jamil Ahmed) who worked accountably to collect data. Lastly, the authors are very much acknowledged to Bangladesh Center for Communication Programs (BCCP) and Institute for Global Tobacco Control, Baltimore, USA. This study was conducted with technical input from the Bangladesh Center for Communication Programs (BCCP) and the Institute for Global Tobacco Control, Baltimore, USA.

## Author Contributions

**Conceptualization:** Saifur Rahman Chowdhury, Tachlima Chowdhury Sunna, Dipak Chandra Das, H. M. Miraz Mahmud, Ahmed Hossain.

**Data curation:** Saifur Rahman Chowdhury, Tachlima Chowdhury Sunna, Dipak Chandra Das, Mahfuzur Rahman Chowdhury, H. M. Miraz Mahmud.

**Formal analysis:** Saifur Rahman Chowdhury, H. M. Miraz Mahmud, Ahmed Hossain.

**Funding acquisition:** Saifur Rahman Chowdhury, Ahmed Hossain.

**Investigation:** Saifur Rahman Chowdhury, Tachlima Chowdhury Sunna, Dipak Chandra Das, Mahfuzur Rahman Chowdhury, H. M. Miraz Mahmud.

**Methodology:** Saifur Rahman Chowdhury, Tachlima Chowdhury Sunna, Dipak Chandra Das, H. M. Miraz Mahmud, Ahmed Hossain.

**Project administration:** Saifur Rahman Chowdhury, Tachlima Chowdhury Sunna, Dipak Chandra Das, Mahfuzur Rahman Chowdhury, H. M. Miraz Mahmud, Ahmed Hossain.

**Resources:** Saifur Rahman Chowdhury, Tachlima Chowdhury Sunna.

**Software:** Saifur Rahman Chowdhury, H. M. Miraz Mahmud, Ahmed Hossain.

**Supervision:** Saifur Rahman Chowdhury, H. M. Miraz Mahmud, Ahmed Hossain.

**Validation:** Saifur Rahman Chowdhury, H. M. Miraz Mahmud, Ahmed Hossain.

**Visualization:** Saifur Rahman Chowdhury, H. M. Miraz Mahmud, Ahmed Hossain.

**Writing – original draft:** Saifur Rahman Chowdhury, Tachlima Chowdhury Sunna, Dipak Chandra Das, Mahfuzur Rahman Chowdhury.

**Writing – review & editing:** Saifur Rahman Chowdhury, Ahmed Hossain.

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
