## [Decision Letter · Decision Letter 0]

2 May 2022

PONE-D-22-07117Compliance with smoke-free legislation in public places: An observational study in a Northeast City of BangladeshPLOS ONE

Dear Dr. Saifur Rahman Chowdhury,

Thank you for submitting your manuscript to PLOS ONE. After careful consideration, we feel that it has merit but does not fully meet PLOS ONE’s publication criteria as it currently stands. Therefore, we invite you to submit a revised version of the manuscript that addresses the points raised during the review process. Our both reviewers have asked for major revisions which have been stated. Kindly resubmit after the clarifications sought.

We look forward to receiving your revised manuscript.

Kind regards,

S. Muhammad Salim Khan

Academic Editor

PLOS ONE

Journal Requirements:

"The authors are very much grateful to Mohammad Shamimul Islam, Team Leader, BCCP Tobacco Control Program, Bangladesh Center for Communication Programs (BCCP), for his continuous supervision, kind cooperation, and encouragement during the entire period of this research project. Heartiest thanks also to Mohammad Shahjahan, Director & CEO, Bangladesh Center for Communication Programs (BCCP), for his continuous cooperation in successfully completing this project. The authors would like to put forward our utmost respect to the authorities of all public places in Sylhet City for their consent and support to collect data. The authors are much grateful to all the data collectors who worked accountably to collect data. Lastly, the authors are very much acknowledged to Bangladesh Center for Communication Programs (BCCP) and Institute for Global Tobacco Control, Baltimore, USA. This study was conducted with technical input from the Bangladesh Center for Communication Programs (BCCP) and Institute for Global Tobacco Control, Baltimore, USA, and financial support from the Bloomberg Initiative."

We note that you have provided funding information. However, funding information should not appear in the Acknowledgments section or other areas of your manuscript. We will only publish funding information present in the Funding Statement section of the online submission form. 

"Support for this study was provided by the Bangladesh Center for Communication Programs (BCCP) with funding awarded by Bloomberg Philanthropies to Johns Hopkins University. Grant number is GC#BCCP/Tobacco Control/2020-57. Saifur Rahman Chowdhury received this grant in reference to the technical and cost proposal from the Bangladesh Center for Communication Programs (https://www.bangladesh-ccp.org/). The content of this publication is solely the responsibility of the authors and do not necessarily represent the official views of Bloomberg Philanthropies or Johns Hopkins University. The funders had no role in study design, data collection and analysis, decision to publish, or preparation of the manuscript."

Reviewers' comments:

Reviewer's Responses to Questions

**Comments to the Author**

1. Is the manuscript technically sound, and do the data support the conclusions?

Reviewer #1: Partly

Reviewer #2: Partly

2. Has the statistical analysis been performed appropriately and rigorously? 

Reviewer #1: No

Reviewer #2: Yes

3. Have the authors made all data underlying the findings in their manuscript fully available?

Reviewer #1: Yes

Reviewer #2: Yes

4. Is the manuscript presented in an intelligible fashion and written in standard English?

Reviewer #1: Yes

Reviewer #2: No

5. Review Comments to the Author

Reviewer #1: The study is conducted with lot of efforts and is being presented effectively. The need of study is also justified. Data are collected with lot of care and is presented well.

There is one objection on analysis which to me seems of a great concern. The average percentages concept used in the analysis, can not be considered correct. Few suggestion for overall percentages

1. Summing up all compliance options with OR condition (for instance if for one case {absent of active smoking =1, presence of no smoking signage = 0, Display of no signage at entrance = 0, no smoking signage is in compliance with law = 1, Absence of cigarretes buts, bidi ends or ashes = 1, absence of smoking aids = 0} for overall compliance it will count "1" for that cases) The overall percentage will be calculated through that column of overall.

this scenario will work if every option is given equal weightage.

2. If every option has different weightage then maximum of the weightage will be the final option.

3. A criterion can be made (e.g. if 4 of the 7 compliances are answered "yes" and 3 are answered "no", overall it will go to "yes"

IF overall percentages are changed by using one of these methods then the manuscript is acceptable in my point of view. If some justification for using average percentages is provided by the author and is acceptable by the editorial board or by some other expert in the field that will be appreciable.

Regards

Reviewer #2: I think the paper addresses an interesting and important research question in a country like Bangladesh with a high current smoking prevalence. However, the English in the present manuscript is not of publication quality for Plos One journal and requires improvement. I recommend the manuscript be copy edited by a native English speaker. In addition, I had the following concerns:

Abstract

• Please indicate the study period in the abstract. Please mention the place of data collection in the objectives part.

• The technique of the data collection should be mentioned in the method section of the abstract. What factors were considered during the observation, (e.g. evidence of smoking (observed smoking, cigarette butt litter, and display of ashtrays, the presence of designated smoking areas/rooms, presence of points of sale or etc. ) and what was the outcome of the study?

• The results section should be limited to the most important numbers. There is no need to bring all detailed results such as % of compliance in each observed section.

Introduction

In the introduction part there is huge information on the burden of smoking worldwide. However, there is lacking on evidence on the smoking prevalence and burden of the tobacco use in Bangladesh.

Methods

Please indicate the reason for the selection of this city to conduct the study. The tool provided by the John Hopkins university is almost a comprehensive tool. Why was it not used alone and researcher decided to modify it through literature review? Which changes have been made to the original instrument? Please describe the changes made in the study tool.

I have some concerns about the operational definitions too. I found them more conceptual rather than operational. Please bring them in a way that reader realize how you defined variables such as active smoking (for example, active smoking in a public place: Active smoking in a public place was marked as present if anyone was seen smoking during the researcher’s visit at the public place being observed for the study) or other variables such as smoking aids, cigarette/bidi stubs/butts, cigarette smell and etc. This part should be rewritten according to the variables included in the tool.

Discussion

Please compare you results with other developing countries or countries close by like Pakistan, India or Nepal. It is better to mention the name of the country instead of the name of the authors for similar studies that actually mentioned in the discussion part. I suggest that at the end of the discussion part the authors elaborate potential solutions and policy options to address low compliance rate. What is your suggestion about the Venue managers? Are they responsible for ensuring that all public places that they operate are smoke free? What can be done to increase population awareness regarding the harm of smoking (social media, tobacco campaign,...)?

6. PLOS authors have the option to publish the peer review history of their article (what does this mean?). If published, this will include your full peer review and any attached files.

Reviewer #1: **Yes: **Muhammad Aasim

Reviewer #2: No

---

## [Author Response · Author response to Decision Letter 0]

15 Jun 2022

We have addressed all the comments raised by the Academic Editor and the Reviewers and reformatted the manuscripts according to the journal guidelines.

---

## [Decision Letter · Decision Letter 1]

29 Aug 2022

PONE-D-22-07117R1

Compliance with smoke-free legislation in public places: An observational study in a Northeast City of Bangladesh

PLOS ONE

Dear Dr. Chowdhury,

Thank you for submitting your manuscript to PLOS ONE. After careful consideration, we have decided that your manuscript does not meet our criteria for publication and must therefore be rejected.

Precisely, the response/rebuttal received against the original reviewers' comments are neither satisfactory nor in compliance with the journal requirements. In all, the revised manuscript fails to address the major concerns raised by the reviewers in their original evaluation of the manuscript.

Under such circumstances we regret that your manuscript cannot be further considered for revision or consideration for publication.

I am sorry that we cannot be more positive on this occasion, but hope that you appreciate the reasons for this decision.

Kind regards,

Koustubh Panda, M. Tech., Ph.D

Academic Editor

PLOS ONE

Reviewers' comments:

Reviewer's Responses to Questions

**Comments to the Author**

1. If the authors have adequately addressed your comments raised in a previous round of review and you feel that this manuscript is now acceptable for publication, you may indicate that here to bypass the “Comments to the Author” section, enter your conflict of interest statement in the “Confidential to Editor” section, and submit your "Accept" recommendation.

Reviewer #1: (No Response)

2. Is the manuscript technically sound, and do the data support the conclusions?

Reviewer #1: Partly

3. Has the statistical analysis been performed appropriately and rigorously? 

Reviewer #1: No

4. Have the authors made all data underlying the findings in their manuscript fully available?

Reviewer #1: Yes

5. Is the manuscript presented in an intelligible fashion and written in standard English?

Reviewer #1: Yes

6. Review Comments to the Author

Reviewer #1: My comments are not addressed.

Either should have corrected the overall compliance percentages or have given a justification of the method of average used.

7. PLOS authors have the option to publish the peer review history of their article (what does this mean?). If published, this will include your full peer review and any attached files.

Reviewer #1: **Yes: **Muhammad Aasim

- - - - -

---

## [Author Response · Author response to Decision Letter 1]

3 Dec 2022

Response to reviewers letter are attached

---

## [Editor Report · Decision Letter 2]

14 Mar 2023

Compliance with smoke-free legislation in public places: An observational study in a Northeast City of Bangladesh

PONE-D-22-07117R2

Dear Dr. Chowdhury,

We’re pleased to inform you that your manuscript has been judged scientifically suitable for publication and will be formally accepted for publication once it meets all outstanding technical requirements.

Kind regards,

Aklilu Habte Hailegebireal, MPH

Academic Editor

PLOS ONE

Additional Editor Comments (optional):

Almost all of the comments and suggestion rised by the reviewers were well addressed and now, i would like to congratulate the authors after declaring that the manuscript is eligible for publication in its current form.
---

## [Editor Report · Acceptance letter]

17 Apr 2023

PONE-D-22-07117R2 

Compliance with smoke-free legislation in public places: An observational study in a Northeast City of Bangladesh 

Dear Dr. Chowdhury:

I'm pleased to inform you that your manuscript has been deemed suitable for publication in PLOS ONE. Congratulations! Your manuscript is now with our production department. 

Kind regards, 

on behalf of

Dr. Aklilu Habte Hailegebireal 

Academic Editor

PLOS ONE